# Interventional therapy of extracranial arteriovenous malformations of the head and neck—A systematic review

Daniel Lilje[1], Martin Wiesmann[1], Dimah Hasan[1], Alexander Riabikin[1], Hani Ridwan[1], Frank Hölzle[2], Omid Nikoubashman[1]*

1 Department of Neuroradiology, University Hospital RWTH Aachen, Aachen, Germany, 2 Department of Oral, Maxillofacial and Facial Plastic Surgery University Hospital RWTH Aachen, Aachen, Germany

These authors contributed equally to this work.
* onikoubashman@ukaachen.de

## Abstract

### Objectives

The primary aim of this study was to conduct a meta-analysis of the literature on interventional treatment for patients with extracranial AVM of the head and neck to identify a superior treatment. The secondary aim was to evaluate the methodological quality of associated articles published between 2000–2020.

### Methods

The literature search was conducted on PubMed, Embase, the Cochrane Library, and scholar.google.com. Studies, meeting the acceptable reference standard underwent meta-analysis. All identified literature underwent methodological quality analysis.

### Results

Of 1560 screened articles, 56 were included in the literature review. Appropriate diagnostic tests were reported in 98% of included articles. 13% of included articles did not specify the embolization agent. Outcome analysis varied throughout. 45% of the authors used radiographic imaging for follow-up. 77% specified the span of follow-up of their entire patient collective. Two articles met the inclusion criteria for meta-analysis. Curing rate of transarterial ethanol embolization for intraosseous AVM was 83% with a complication rate of 58%. Curing rate of ethanol combined with NBCA or Onyx in soft tissue AVM was 18% with a complication rate of 87%.

### Conclusion

Our literature review revealed an absence of treatment or reporting standards for extracranial AVM of the head and neck. The meta-analysis is comprised of two articles and methodological quality is heterogeneous. We recommend implementing consistent reporting standards to facilitate comparability of studies and to provide robust data for the development of an evidence-based treatment strategy.

**Data Availability Statement:** All relevant data are within the paper and its Supporting information files.

**Funding:** The author received no specific funding for this work.

**Competing interests:** The authors have declared that no competing interests exist.

## Advances in knowledge

Meta-analysis showed a favorable radiological outcome for intraosseous AVM when treated with intraarterial ethanol embolization. Our analysis demonstrated that the published data on extracranial AVMs of the head and neck is lacking in consistency and quality, prompting agreement for the need of standardized reporting on AVM treatments.

## Introduction

Cutaneous vascular anomalies are rare conditions that can be very demanding for treating clinicians. The spectrum of cutaneous vascular malformations includes high-flow lesions such as arteriovenous malformations (AVM) and low-flow lesions such as venous malformations (VM) and lymphatic malformations (LM). AVM, which are discussed in this work, present with a wide range of symptoms and variable progression. Misdiagnosis and recurrence are not uncommon [1] and can lead to life-threatening situation with serious hemorrhages [2] or congestive heart failure [3]. Additionally, inconsistent nomenclatures [4] and treatment modalities lend confusion in deciding on an approach for resolution or palliation of these lesions that may considerably limit the patients' quality of life [5].

In 2018, the International Society for the Study of Vascular Anomalies (ISSVA) released a comprehensive classification catalogue [6] subdividing vascular malformations into simple and combined capillary, lymphatic, venous and arteriovenous malformations, according to their vascular architecture (Table 1). AVMs are characterized by an abnormal vascular network directly connecting arteries and veins to form a fast flow lesion that lacks a capillary bed [7].

In this review we focused on the analysis of articles covering the treatment of extracranial AVMs of the head and neck with the use of sclerotherapy and embolization. An illustrative example of an intraarterial embolization procedure is given in Fig 1.

Prior to the initial research, we developed the inclusion criteria for eligibility for meta-analysis (or acceptable reference standard). Methodological quality standards for analyzing the remaining articles were developed during the review process.

We address caregivers of all stages of experience and all disciplines managing these vascular malformations, with the goal of better understanding the results of the different AVM management methods used over the last two decades. Using the PICO approach, we performed a

**Table 1. Overview of ISSVA-Classification for vascular malformations.**

| Vascular Tumors | Vascular Malformations | |
|---|---|---|
| | Simple | Combined |
| • Benign <br> • Locally (aggressive/ borderline) <br> • Malignant | • Capillary Malformations <br> • Lymphatic Malformations <br> • Venous Malformations <br> • Arteriovenous Malformations <br> • Arteriovenous Fistula | CVM, CLM <br> LVM, CLVM <br> CAVM <br> CLAVM <br> other |

CVM = Capillary-Venous-Malformation, CLM = Capillary Lymphatic Malformation, LVM = Lymphatic Venous Malformation, CLVM = Capillary Lymphatic Venous Malformation, CAVM = Capillary Arteriovenous Malformation, CLAVM = Capillary Lymphatic.

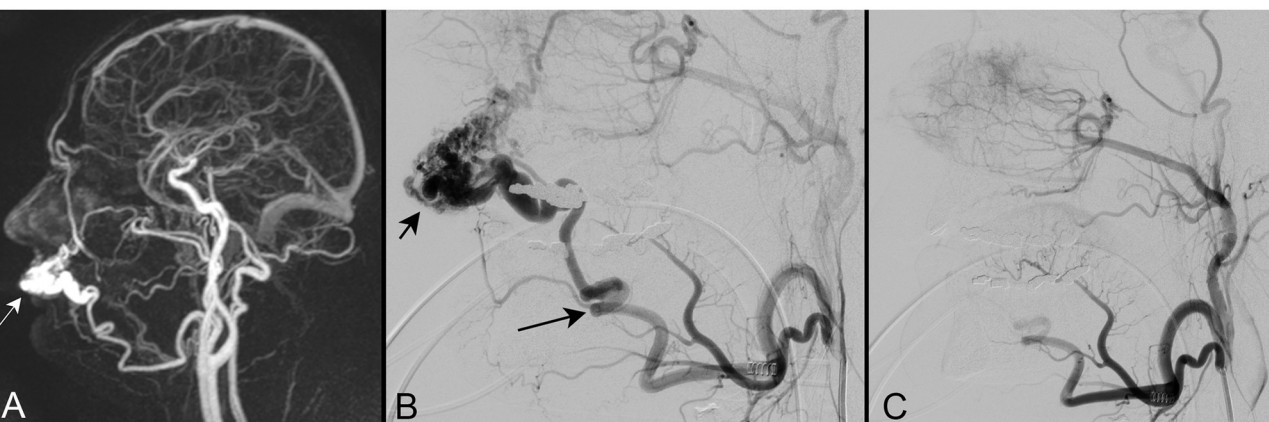

**Fig 1. Illustrative example of 4D-MR angiography in patient with extracranial AVM.** Patient in his forties with a progressive pulsating mass on his left upper lip. Magnetic resonance imaging using a dynamic 4D-MR angiography sequence (Fig 1A, sagittal reconstruction in mixed phase) reveals a dense vascular mass (arrow) on the upper lip with arteriovenous shunts. Diagnosis of arteriovenous malformation (AVM) was established. Catheter angiography with selective injection of the left external carotid artery (Fig 1B, lateral view) confirmed the diagnosis. The AVM (short arrow) was fed through the facial artery (long arrow) and drained predominantly into the facial vein. The facial artery was selectively catheterized using a microcatheter and embolized with microparticles (150–250 µm). At the end of the embolization procedure (Fig 1C, selective injection of the left external carotid artery in lateral view) the AVM is completely eliminated. The other branches of the external carotid artery are preserved.

systematic review of relevant literature published between 2000 and 2020 to assess these interventions and their results [8].

## Methods

### PICO database search

**Population.** In March 2020 we identified suitable studies using the online search engines PubMed, Embase, the Cochrane Library and scholar.google.com. All clinical studies targeting diagnosis, treatment, outcome, complication rate and recurrence of AVMs, venous malformations, and lymphatic malformations, as well as mixed lesions of the head and neck were included in the primary review [6]. Further publications identified in citations and quotes throughout the initial literature review were included in scope and underwent the same review process, with the last addition made in August 2020. Retrospective and prospective English studies, as well as case reports published between January 2000 and March 2020 were included. This studied timeline marks a rapid progression in the evolution of interventional radiology. There were no restrictions as to the country of origin, clinical setting or size of the institution in which the treatments were performed. We did not set a minimum sample size of the studies, as any clinical findings or experience in treatment might be of value to clinicians or future studies. Management of capillary malformations, vascular tumors, and vascular malformations associated with syndromic diseases were not discussed in this literature review.

For a broad overview, we searched various English medical terms for the description of vascular malformations, their subclassification and possible anatomical sites of interest identified in previous database searches and previewed articles. Medical subject headings (MeSH) items were added to the database research, accordingly. We intentionally included obsolete search terms (such as "hemangiomas") to ensure inclusion of older publications [4]. A full disclosure of the search terms used, according to the PICO strategy, is shown in S1 Table. The full search string for PubMed is enclosed in S1 Appendix.

Only publications targeting AVMs of the head and neck region underwent quality analysis, irrespective of their anatomical location (intramuscular, intraosseous, and subcutaneous). As

details on gender, age, and ethnic background of the treated patients were inconsistently reported, they did not undergo analysis.

**Intervention.** We focused on treatment of soft tissue AVMs of the head and neck with sclerotherapy and embolization as primary therapy. This includes interventions with any applied sclerosant agent and either pharmaceutical embolization or mechanical embolization with coils, balloons, and vascular plugs through direct lesion puncture or endovascular trans-arterial or transvenous approach. Where sclerotherapy or embolization were part of the treatment scheme, studies targeting surgical resection were also included in the primary search. To ensure we did not overlook any alternative methods and treatment options, we expanded the search with general MeSH terms and items for any other sclerotherapeutic intervention, and did not specify substance and generic names for utilized agents in our search.

**Comparison.** Leveraging meta-analysis, we compared radiological outcome and complication rates of the different treatment options for extracranial AVMs with the goal of identifying a superior treatment method within the existing literature.

**Outcome.** We included publications that documented the progression of treated patients in terms of resolution, palliation, and recurrence measured by clinical inspection, radiological imaging, and subjective patients' or physicians' description. The database search also included any aesthetic results and quality of life records conducted by questionnaires or interviews.

## Qualitative analysis of methodology

All initial diagnostic tests evaluating the vascular architecture of the lesion were deemed to be of sufficient quality if they were applied in all patients, regardless of whether or not criteria for the choice of method was defined. We considered angiographic imaging to be the gold standard.

To be considered for inclusion, the outcome for each patient had to be measured by comparable tests with reproducible results and re-test reliability.

The post-treatment follow-up was deemed of adequate quality if it was reported for all individuals in the study, with the ideal follow-up time spanning one year between evaluation and the last treatment session [9]. Additionally, description of symptoms and location, as well as details on complications that arose had to be comprehensively reported.

Instrument-based assessment and analysis of the risk of bias were planned but not conducted, since the analyzed studies were retrospective, non-randomized, non-comparative clinical trials and case studies and thereby hold the inherent potential for confounding, selection bias, reporting bias as well as information bias [10].

## Inclusion criteria for meta-analysis (Acceptable reference standard)

**1) Quality of description of diagnostic measures.** Only studies that used angiographic imaging (e.g., magnetic resonance imaging (MRI), computed tomography (CT), digital subtraction angiography (DSA) or fluoroscopic angiography) were included. Ultrasonography was excluded due to inconsistent presentation of AVMs and user-dependence of ultrasound examinations [11]. Clinical examination and clinical history alone were considered insufficient for a reliable diagnosis.

**2) Quality of description of therapy.** All publications covering sclerotherapy and/or embolization with and without surgical interventions on extracranial AVM of the head and neck were accepted. Information on the sclerosant or embolic agent had to be given.

**3) Quality of description of results.** To eliminate the risk of a group classification error, we only included studies that defined cure or resolution as devascularization rates of more than 99% compared to baseline. Hence, angiographic imaging had to be performed before and after the intervention to demonstrate the results of the treatment.

**4) Quality of follow-up.**    For cerebral AVM, a minimum follow-up period of one year is recommended to detect lesion recurrence after therapy [9]. Analogously, a follow-up time of at least one year was considered necessary for inclusion in our meta-analysis. As mean or median time of follow-up of the study group (as well as maximum) time do not yield information on the individual minimum follow-up time, that data alone was considered insufficient, and the related literature was excluded from analysis.

## Statistical analysis

For statistical analysis we used cross-table and Pearson's Chi-Square calculations. All statistical analyses were performed with SPSS 25 software (IBM, Armonk, NY).

# Results

## Overview

Using the selection process outlined in Fig 2, we identified 1520 articles that went into primary title and abstract review, 151 of which fulfilled the criteria for our full text review. Screening their references yielded another 7 publications, for the total of 158. Of the total articles identified, 33 addressed AVMs. An additional search made using the PICO strategy without the link between items identified a further 23 articles, resulting in a total of 56 articles to qualify for full text review. The full text review of 56 articles identified 910 patients. Of the articles reviewed, 52% (29/56) were case reports or case series. Only two articles with a total of 57 patients stood up to our inclusion criteria for meta-analysis [12,13], A summary of the qualitative synthesis of the methodology of items of all 56 articles is given in S2 Table.

The summary represents the authors' conclusion about the quality of each item.

A comprehensive overview of each article is provided in the supplemental S3 Table. S2 Appendix shows the full list of excluded studies on full article assessment level.

## Quality of description of diagnostic measures

All but one of the 56 publications reported an appropriate diagnostic test for initial diagnosis. Nine authors described using only one singular imaging modality to confirm the clinically discovered type of lesion; all other articles reported two or more imaging modalities for diagnosis. Details of the imaging modalities are outlined in Table 2.

## Quality of description of therapy

All 56 authors used sclerotherapy or embolization as part of the therapy plan. Twenty-seven used sclerosant agents and/or pharmaceutical embolization without surgical interventions. (Table 3).

The remaining 29 studies included a surgical procedure as part of their treatment plan, either compulsory, optional, or as an escalation of treatment.

Seven out of 56 authors (13%) did not specify the pharmaceutical agent used [14–20].

Overall, there is a noticeable disparity of treatment algorithms. For instance, Kansy et al. [21] showed a clear treatment algorithm which showed the indication for the therapy applied. All other authors lacked an explanation for their individual treatment decision.

## Quality of description of results

Pre- and post-treatment imaging analysis were deemed the measurement of choice for evaluation of therapy outcome, with 45% (25/56) of authors reporting imaging parameters for outcome analysis. Stratification of outcome ranged as shown in Table 4.

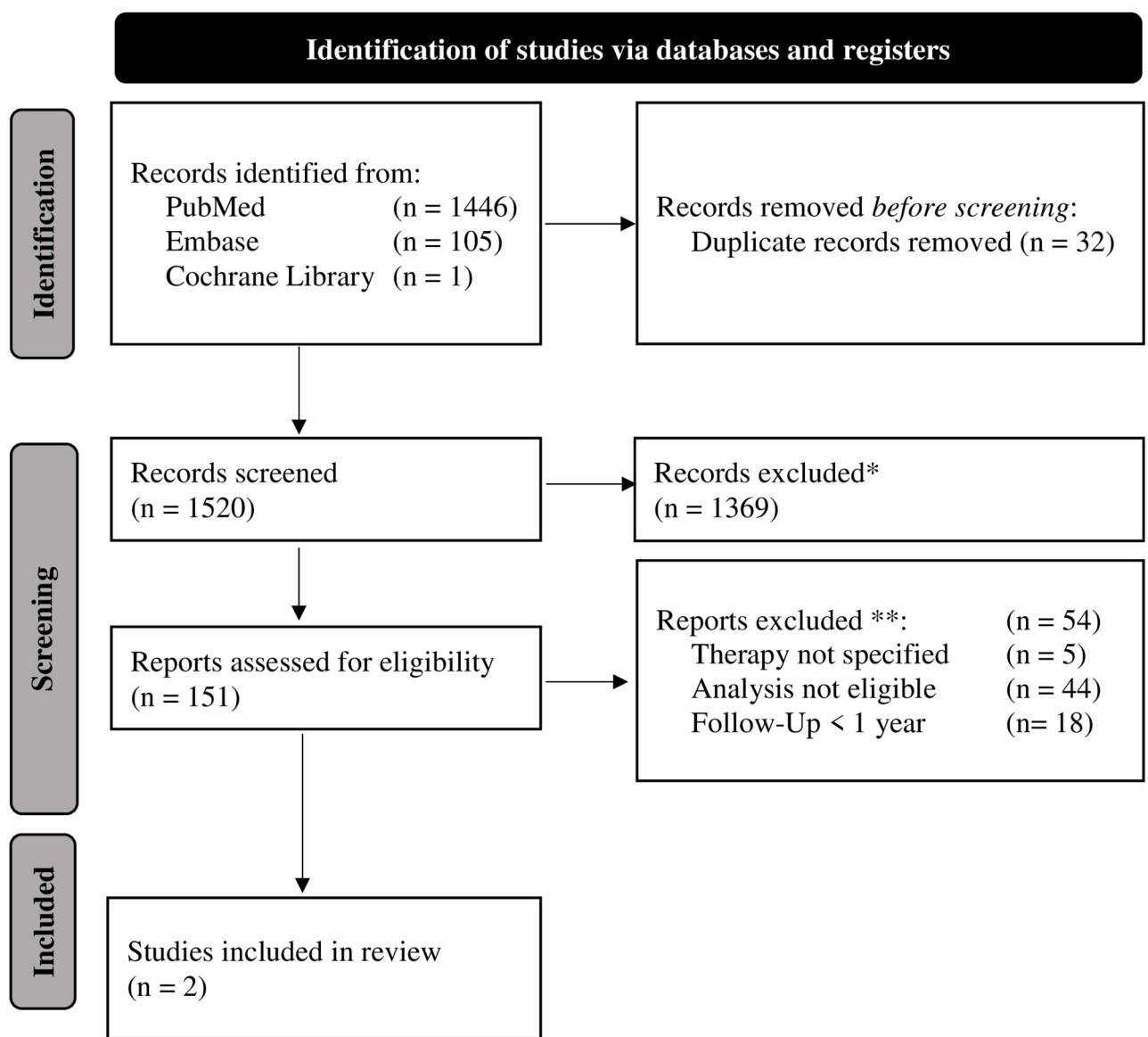

**Fig 2. PRISMA flow diagram.** * Exclusion if not: AVM of head and neck, English language, published 2000–2020, Interventional treatment applied. ** More than one may apply.

**Table 2. Imaging modalities used.**

| Imaging modality | Ratio of modality usage |
|---|---|
| Conventional angiography | 30/56 (53.6%) |
| Computed tomography | 29/56 (51.8%) |
| Magnetic resonance imaging | 28/56 (50.0%) |
| Digital Subtraction Angiography | 9/56 (16.1%) |
| Doppler-Ultrasound | 5/56 (8.9%) |
| Other * | 2/56 (3.6%) |

* Scintigraphy and Time-Of-Flight-MRI.

**Table 3. Ratio of therapeutic agents used.**

| Most used Embolization Agent | Number of Authors to use the agent |
|---|---|
| N-Butylcyanoacrylate | 20/48 (42%) |
| Polyvenylalcohol | 15/48 (31%) |
| Coils | 13/48 (27%) |
| Ethanol | 10/48 (21%) |
| Onyx | 8/48 (17%) |
| Bleomycin | 4/48 (8%) |

Articles may have reported multiple agents.

Some authors refrained from using quantitative stratification for their evaluation and instead used wording such as 'notable reduction' (Chelliah et al. [15]), 'no newly developed nidus' (Kim et al. [13]), or 'retention within the vascular lesion' (Hsiao et al. [32]). Also, descriptions of clinical outcomes were limited to the physicians' examination and patient report and included domains such as 'no recurrence' (Shobeirian et al. [33]), 'stable condition' (Ermer et al. [34]), 'reduction in size' (Fujita et al. [35]) or 'symptoms were found to subside' (Ishimaru et al. [35]). Richter et al. [36] used an interview to evaluate the quality of life of the patients before and after the intervention. A standardized questionnaire, however, was not provided. Kaji et al. [37] used visual evaluation of photographs and the appearance of the lesions alongside a volumetric response calculation based on MRI taken before and after treatment. [38] Saito et al. [39] described their findings to be the 'desired result' in cosmetic appearance. Gegenava et al. [39] used a telephone interview to ask for any recurrent bleeding episodes. In studies that addressed multiple types of vascular lesions (AVM, LM, VM) (Churojana [39]) or multiple locations of the AVM (head, neck, trunk, limbs) (Kitagawa [39]), it was not possible to differentiate the outcome for each type of lesion.

## Quality of follow-up

Forty-three of 56 articles (77%) reported the time of minimum follow-up for their whole patient collective, averaging to 488.5 days (range 7 days to 2880 days, SD 577.1 days). The remaining 13 articles did not include adequate follow-up data. One article documented 'ongoing' follow-up while the remaining four articles reported either mean or maximum follow-up time, with no mention of individual timelines. Eight authors recorded no follow-up information whatsoever.

## Meta-analysis

Two articles fulfilled our inclusion criteria for meta-analysis.

**Table 4. Outcome stratification.**

| Authors | Defined devascularization rate for 'cure' |
|---|---|
| Kitagawa et al. [22], Su et al. [12], Wang et al. [23], Han et al. [24], Pompa et al. [25,26], Fan et al. [27], Bhandari et al. [28], Zheng et al. [29] | 100% |
| Kim et al. [13] | >99% |
| Meila et al. [14], Gupta et al [30], Zhao et al. [31] | >90% |

Defined devascularization rate (%) authors refer to as 'cure' (best outcome possible). The remaining authors did not use devascularization as measurement of outcome.

**Table 5. Overview of meta-analysis.**

|  | Su et al. [12] | Kim et al. [13] | P-Value |
|---|---|---|---|
| n Patients | 12 | 45 | - |
| Resolution | 83% | 18% | <0.001 |
| Mean sessions | 1.25 | 2.93 | 0.004 (r = 0.38) |
| Failure | 0% | 6.7% | 0.358 |
| Complications | 58.3% | 86.7% | 0.027 |

r = medium effect size.

In the first study, Su et al. [12] have described the management of 12 patients with AVMs of the maxilla and mandible, admitted with a history of recurrent hemorrhaging and an acute bleeding episode with the need for emergency treatment. Using direct puncture and transarterial ethanol embolization with intravenously deployed coils, they had a curing rate (defined as 100% devascularization of the nidus) of 83% (10/12 patients). Two patients had partial remission and were waiting for additional treatment. Patients went through a mean number of 1.25 treatment sessions (range 1 to 2 sessions).

In the second study, Kim et al. [13] reported the interventional treatment of 45 patients with soft tissue AVMs in the head and neck region using transarterial or direct-puncture ethanol sclerotherapy, and additional surgical resection in 23 of the 45 patients. NBCA (N-Butyl Cyanoacrylate), Onyx (ev3 Neurovascular, Irvine, California, USA) or coils were used as adjunctive material. The AVM was completely eradicated in 18% (8/45) of patients (confirmed in angiography). Their patients were treated 2.93 times on average (range 1 to 9 sessions).

Complete resolution, defined as 100% devascularization of the AVM, was significantly more likely in the series of Su et al. [12], who used ethanol with coils in intraosseous AVM of the jaw, compared to ethanol along with NBCA or Onyx in soft tissue AVM in the series of Kim et al. [13] (p<0.001). There was no significant difference concerning treatment failure, defined as devascularization below 50% of the baseline with failure rates of 0% and 6.7% in the series of Su et al. [12] and Kim et al. [13] respectively (p = 0.358).

Overall complication rates differed significantly between the two treatment approaches (p = 0.006). Using Ethanol alone, Su et al. [12] reported complications in 50% of cases. Using a combination of Ethanol and NBCA or Onyx, Kim et al. [13] reported complications in 87% of cases. There, too, was a significant difference between the mean amount of treatment sessions each patient went through. Patients in the study of Su et al. [12] underwent 1.25 sessions, versus 2.93 sessions in the study of Kim et al. [40] (p = 0.004). Considering the small number of studies, we refrained from using visualization of the meta-analysis. The overview of the meta-analysis is outlined in Table 5.

## Discussion

A major finding of our analysis was that therapeutic approaches for the management of AVMs vary considerably, not yielding treatment or reporting standards. Our literature review identified 56 articles with sclerotherapy and embolization as the primary treatment modality for AVMs of the head and neck. None of these used the same approach. Remarkably, none of these studies was prospective, hinting towards a publication bias. To our surprise, only two articles fulfilled the acceptable reference standard for meta-analysis. Hence, we were not able to reach our primary aim: to conduct a meta-analysis of interventional treatment for patients with extracranial AVM of the head and neck to identify a superior treatment.

The two articles included in the meta-analysis used ethanol applied by direct puncture of the nidus and transarterial injection (Su et al. [12]), or both direct nidus injection and transarterial application of ethanol with adjuvant use of either NBCA or Onyx (Kim et al. [13]). The approach of Su et al. [12] in the treatment of intraosseous AVMs of the jaw yielded a higher chance for beneficial outcome. They reached complete devascularization in 83% of patients which is superior to 18% of patients in the series of Kim et al. [13], who treated soft tissue AVMs (p<0.001). Kim et al. [13] performed additional surgical resection in 23 of 45 cases to correct residual cosmetic problems or to remove residual fistulas. In addition, Su et al. [12] needed significantly fewer treatment sessions than Kim et al. [13] (1.25 sessions versus 2.93 sessions) (p = 0.004) and had a significantly lower complication rate (50% versus 87%) (p = 0.006).

Given the low number of included cases, the varying reporting standards, and the heterogenous cohorts, the results of our qualitative analysis of the methodology must be interpreted with great care. For instance, the reviewed articles (excluding case series and case reports) included varying numbers of pre-treated patients with a range of 10% (6/62 patients)(33) to 90% (9/10 patients) [36]. Only 52 articles provided details on lesion extent and location, with varying accuracy. For example, Kim et al. [13] used the Schobinger system [41] to describe the characteristics of the AVM but did not report on the exact location in the head and neck region. Most articles summarized the location of the lesions with macroscopic expressions such as 'cheek', 'temporal', or 'forehead'. Zou et al. [42] noted a difference between 'localized' or 'diffuse' lesions, depending on the tissue planes through which the AVM extended. AVM location, however, may have an impact on outcome rates, as has been suggested by Chen et al. [43], who found that intraosseous AVM of the jaw had a higher rate of complete involution in comparison to soft tissue lesions (89.3% and 60.7% respectively). With regard to this, Su et al. [12] reported of intraosseous AVM, which makes it questionable whether management and outcome of these AVM can be compared to that of cutaneous or soft tissue AVMs as studied by Kim et al. [13].

Furthermore, analysis of the results is impeded by the variety of treatment approaches and their combinations. Approaches to AVM obliteration include percutaneously and intravascularly applied sclerosants such as Ethanol, STS (Sodium Tetradecyl Sulfate), OK432 (Picibanil), and Bleomycin. These agents stimulate local inflammation of the endothelia causing thrombosis which occlude the vessel [44–46]. The risk of systemic or distal reactions to the agent has to be thoroughly assessed. Additionally, applied agents (NBCA, Polyvenylalcohol (Onyx), or galantine microspheres) block collaterals or outflow veins of the nidus [47]. Mechanical occlusion with coil depletion into the nidus is also reported [24]. In AVMs with multiple feeding arteries and a single draining vessel, direct puncture embolization may result in a more favorable outcome than the transarterial approach [48]. Adjuvant surgical interventions may be necessary to facilitate cosmetic reconstruction or restore essential structures.

Aside from interventional treatment options, targeted drug therapy has been the subject of recent research [49]. With vascular anomalies showing somatic mutations, cellular signaling pathway inhibitors such as Sirolimus may have a positive effect [50].

The side effects and complications arising from the use of sclerosants and embolization agents has been well-reported [51–56]. Forty-three of the 56 articles included a description of the complications arising during or after the procedures. Considering all reported complications, the mean complication rate amongst all articles, including case reports, was 31% -ranging from 0% to 100% (SD 36.8%). Complications were, however, inconsistently described. One author presented complications according to the Society of Interventional Radiology (SIR Table 6) [57] but did not provide any further details on the nature or management of the adverse effects.[58] Some implemented their own categorization, similar to the SIR, and gave

**Table 6. Society of Interventional Radiology (SIR) classification system for complications and outcome.**

| Minor Complications | |
|---|---|
| A. | No therapy, no consequence |
| B. | Nominal therapy, no consequence; includes overnight admission for observation only. |
| Major Complications | |
| C. | Require therapy, minor hospitalization (<48 hours) |
| D. | Require major therapy, unplanned increase in level of care, prolonged hospitalization (>48 hours) |
| E. | Permanent adverse squelae |
| F. | Death |

further details on presentation and management [12,35]. Several authors did provide a statement as to whether or not complications had occurred, but did not specify what the complications were [13,22,59]. All others provided a detailed description of adverse effects and the management.

When looking at the complications in our meta-analysis group, we identified a significant difference between the occurrence of adverse effects. The high complication rate of 50% (Su et al. [12]) and 86.7% (Kim et al. [13]) in the usage of ethanol is, however, consistent with existing literature. Ethanol is known to cause necrosis, skin ulceration, nerve damage [60] and hemoglobinuria [61] as well as pulmonary hypertension if serum levels are too high [55]. Amongst the 56 articles, minor and reversible treatment side effects were extensively reported. They included skin ulcerations, discoloration, pain, and transient swelling and were conservatively managed. Major complications were reported by Kim et al. [13] and Pekkola et al. [59] and included swelling with the need for surgical decompression, one cerebral infarction and one retinal ischemia. In one study the patient died due to soft tissue necrosis and septicemia [62].

Lastly, varying outcome assessments and definitions make comparisons difficult. For instance, a devascularization above 90% is considered "complete" by three authors, but only considered as "partial" by nine others (Table 4). Additionally, non-standardized subjective outcome parameters impair comparability. In their systematic review, Horbach et al. [63] analyzed the quality of instruments for used for peripheral vascular lesions in patient- and physician-reported outcomes. They found few reliable and adequate measurement instruments that reflect the outcome of therapeutic interventions. They suggested that new specific instruments need to be developed to cover all relevant outcome domains. However, their work did not review the quality of radiologic assessment of vascular malformations.

Overall, research on treatment options for peripheral AVMs lack prospective, randomized trials and the reporting standards necessary to develop evidence-based management of these complex lesions. Therefore, we suggest the implementation of reporting standards for publications that include 1) a dynamic and/or angiographic imaging modality confirming the AVM and distinguishing it from other vascular entities, 2) a detailed description of the therapeutic management, 3) the definition of a devascularization of greater than 99% for complete resolution 4) a follow-up time of at least one year, with post-treatment imaging.

## Strengths and limitations

This literature review has its strengths, particularly in the methods. The literature research was conducted in a very broad manner and articles on all vascular lesions, including AVM as well as VM and LM, were included in the initial search. After primary literature review, further relevant references were identified, adding to the integrity of the literature collection. The

research was conducted using publicly open sources and is therefore reproducible. There are, however, several limitations. The initial literature research was conducted by only one individual which poses a risk of biases and errors. This risk was minimized by thorough assessment and consultation of experienced authors of medical publications. The inclusion criteria for meta-analysis were decided on within our institution, potentially leading to reporting and confirmation bias. Moreover, the identified articles focus on different localizations of AVM in the body, like intraosseous or soft tissue lesions. This leads to a heterogenic sample group which makes generalization difficult. Finally, some of the studies have a small sample size or are case reports and may therefore not be representative of the target population.

## Conclusion

Our meta-analysis implies that intraarterial ethanol embolization result in high rates of complete devascularization of extracranial AVMs of the head and neck located in the bone tissue. However, our qualitative literature analysis revealed that only a small number of articles fulfil a level of methodological quality that allows comparison of treatment effects. Current literature is missing a standardized reporting system for diagnosis, management, and outcome of extracranial AVM, impairing comparability. Despite a considerably large number of publications on the topic, we conclude that to-date there is no robust evidence for a superior treatment strategy. Therefore, we recommend the implementation of standardized criteria for publications that include 1) a dynamic and/or angiographic imaging modality confirming the AVM and distinguishing it from other vascular entities, 2) a detailed description of the therapeutic management, 3) the definition of a devascularization of greater than 99% for complete resolution 4) a follow-up time of at least one year with post-treatment imaging.

## Supporting information

**S1 Checklist.**
(DOC)

**S1 Table. PICO search terms.** Complete list of terms used for database search.
(DOCX)

**S2 Table. Summary of the qualitative synthesis of methodology.**
(DOCX)

**S3 Table. Overview of meta-analysed studies: Su et al. [12].**
(DOCX)

**S4 Table. Overview of meta-analysed studies: Kim et al. [13].**
(DOCX)

**S5 Table. Comprehensive overview of reviewed articles not eligible for meta-analysis.**
(DOCX)

**S1 Appendix. Complete PubMed search string.**
(DOCX)

**S2 Appendix. Complete list of excluded studies.**
(DOCX)

## Author Contributions

**Conceptualization:** Daniel Lilje, Martin Wiesmann, Omid Nikoubashman.

**Formal analysis:** Daniel Lilje.

**Investigation:** Daniel Lilje, Omid Nikoubashman.

**Methodology:** Daniel Lilje, Martin Wiesmann.

**Project administration:** Martin Wiesmann.

**Supervision:** Martin Wiesmann, Omid Nikoubashman.

**Validation:** Omid Nikoubashman.

**Visualization:** Daniel Lilje.

**Writing – original draft:** Daniel Lilje.

**Writing – review & editing:** Martin Wiesmann, Dimah Hasan, Alexander Riabikin, Hani Ridwan, Frank Hölzle, Omid Nikoubashman.

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
