## [Decision Letter · Decision Letter 0]

29 Dec 2021

PONE-D-21-11556Interventional therapy of extracranial arteriovenous malformations of the head and neck – A systematic reviewPLOS ONE

Dear Dr. Nikoubashman,

Thank you for submitting your manuscript to PLOS ONE. After careful consideration, we feel that it has merit but does not fully meet PLOS ONE’s publication criteria as it currently stands. Therefore, we invite you to submit a revised version of the manuscript that addresses the points raised during the review process.

 The topic of your article maybe of interest for a Neurointerventional readership. However, Reviewer 1 raised major concerns on the methodology of the meta-analysis including the use of adequate statistics. Please seek professional statistical and methodological support in performing a valid meta-analysis of the presented literature including search strategy, results, quality and bias assessment of the included literature and discussion! Please address all concerns carefully in your revision! 

We look forward to receiving your revised manuscript.

Kind regards,

Stephan Meckel, MD, PhD

Academic Editor

PLOS ONE

Journal Requirements:

2. Please use an established tool for the quality assessment of non-randomized interventional studies (e.g. the ROBINS-I https://www.bmj.com/content/355/bmj.i4919), which is also recommended in the reference #10 cited in your manuscript. Thank you for your attention to this request.

Reviewers' comments:

Reviewer's Responses to Questions

**Comments to the Author**

1. Is the manuscript technically sound, and do the data support the conclusions?

Reviewer #1: Partly

2. Has the statistical analysis been performed appropriately and rigorously? 

Reviewer #1: No

3. Have the authors made all data underlying the findings in their manuscript fully available?

Reviewer #1: Yes

4. Is the manuscript presented in an intelligible fashion and written in standard English?

Reviewer #1: No

5. Review Comments to the Author

Reviewer #1: Overall

Thank you for all of your efforts in this work. There is a clear rationale and introduction of the void in our knowledge of AVMs. The writer is up to date in technical and anatomic jargon of the topic. Most tables and supporting data are complete and of value.

I do have major concerns in the methodological set up of the paper. Throughout the review different objectives are stated and it seems if the main goal was altered in the process of writing the manuscript. In addition various adapted risk of bias tools are available for non-RCT studies, these could be easily applied to the present review. Although the writers are critical and aware of the potential biases in the selected articles and opted to critically appraise in Table S2, I believe among others study design and sample size should be included (e.g. assessing the papers on directness of evidence and risk of bias). At this point the review lacks an easily assessable overview (i.e. flowchart or supporting figure) of the various treatments performed and its outcome in terms of for example success or complication rate, which I believe is the main objective of the review. Plenty of published reviews have coped with heterogeneity in results and alternatively analysed (e.g. semi-quantitative) and visualized these results in various supporting manners.

I would also strongly advise to re-write the paper with a native English speaker as the paper contains multiple errors which decreases the joy of reading.

Abstract

Objective: I believe the secondary aim is obsolete as a methodological assessment of the selected papers is one of the key elements of performing a systematic review.

Methods: “Criteria for meta-analysis were implemented.” I think you are meaning to say that a meta-analysis was performed?

Results: “Outcome analysis 38 varied with 45% of the authors using imaging parameters and 77% indicating the span of 39 follow-up of their entire patient collective” > this sentence is very vague for a new reader who was not involved in the research process, please re-write.

Conclusion: This conclusion is not clear from the results section, please align these paragraphs.

Advances in knowledge: Please add “data on AVMs /reporting standards addressing treatment of AVMs”, in this manner it seems you are referring to all data available.

Introduction:

Lines 69 -70: This objective is different of the objective in the abstract, even though outcome is a consequence of a treatment, you should define what is “most effective treatment” ? What outcome are you interested in?

Lines 70-71: See previous comment: evaluating the methodological quality of these papers is part of the systematic review.

Line 73: Why are you only addressing the papers from 2000? Perhaps due to interventional developments? Please add your rationale.

Methods

Lines 90-92: Consider to only mention your search in August 2020, it is OK to update your search once it gets outdated during the writing process, though only if it is performed in a similar fashion. I would recommend to add a second reviewer in the primary selection of eligible articles (or a substantial percentage of the eligible articles)

Lines 112-113: In this statement you are narrowing your research question and this should be used throughout your manuscript.

Line 123: To my opinion comparison in the PICO structure should be the other invention group you are comparing with, so that would be any primary treatment other than “sclerotherapy and embolization” or any patient in need of a secondary treatment and so on. At this point the paragraph after “Comparison” is just a brief sentence of your “Outcome” paragraph and does not give any additional information.

Line 142: The authors should seek to another approach to adequately assess directness of evidence and risk of bias in the selected articles. They could also consider to discuss the case reports separately from the other articles with a sample size of e.g. >15 or >20 patients.

Results

Line 252: “four articles reported either mean of maximum follow-up time only” > if this FU was longer than 1 year I guess these articles could still be included (as is stated in the methods). If so, please add why these papers were excluded.

For a meta-analysis, Pearson Chi square and Mann Whitney U tests are not appropriate.

Discussion

Another goal “representative meta-analysis” is added for the first time in the discussion part. The objectives should be similar throughout the manuscript.

The discussion shows that the authors do have substantial knowledge of the subject as they are able to discuss and reflect on the different treatment options. It would be of interest if the authors could add their view on what the suggested reporting standards should contain or focus on.

Nevertheless all information on the meta-analysis is hazardous to me, since I doubt if the authors have used the appropriate statistical approach.

Line 367-369: To my opinion a long list of Author et al is not reader-friendly, please re-write.

Conclusion

All recommendations for reporting standards should be placed in the discussion. In addition please explain why >99% (why not 90%?) devascularisation is used and how this should be measured.

Also the strengths and limitations should be placed in the discussion.

Fig 1 PRISMA flowchart

- Please take note of the “simple” PRISMA set up of the flowchart in other published systematic reviews. This flowchart is quite chaotic and hard to follow for an interested reader.

Supporting data:

- Please add all supplemental data to one document to help the reader find the information they need to comprehend the message of your paper

- It would be better to give an summary score / rating to the separate articles in the table S2 in order to see which articles are the most appropriate to answer your research question. Also the table should stand on its own, meaning the table should be informative enough to fully understand independently of the paper. In my opinion the legend is too brief for a reader which is not as familiar with the subject as the writers.

- I understand the length of the Table S5 makes it unfavorable to add in the main body of the text, but as this table may be the most important finding of the study I would recommend the writers to think of another way to add this information to the main manuscript. Perhaps a clean flowchart with roughly categorized treatments and their success or complication rate would be of interest.

6. PLOS authors have the option to publish the peer review history of their article (what does this mean?). If published, this will include your full peer review and any attached files.

Reviewer #1: No

---

## [Author Response · Author response to Decision Letter 0]

12 Feb 2022

Review Notes

Note #1

I do have major concerns in the methodological set up of the paper. Throughout the review different objectives are stated and it seems if the main goal was altered in the process of writing the manuscript.

Response: We would like to thank the reviewer as he/she accurately points out that in writing the manuscript, the description of the objects of the review are stated in a varying fashion. We have identified the inaccuracy in the text and changed the phrasing accordingly.

The following section was changed in the Abstract section:

“The primary aim of this study was to conduct a meta-analysis of the literature on interventional treatment for patients with extracranial AVM of the head and neck to identify a superior treatment. The secondary aim was to evaluate the methodological quality of associated articles published between 2000 - 2020.” (Clean Copy Line 25-28)

The following paragraph was changed in the Introduction:

“In this review we focused on the analysis of articles covering the treatment of extracranial AVMs of the head and neck with the use of sclerotherapy and embolization”. (Clean Copy Line 74-75)

The following sections was changed in the Discussion:

“A major finding of our analysis was that therapeutic approaches for the management of AVMs vary considerably, not yielding treatment or reporting standards.” (Clean Copy Line 303-304)

“Given the low number of included cases, the varying reporting standards, and the heterogenous cohorts, the results of our qualitative analysis of the methodology must be interpreted with great care.” (Clean Copy Line 324-326)

The following section was changed in the Conclusion:

“Our meta-analysis implies that intraarterial ethanol embolization result in high rates of complete devascularization of extracranial AVMs of the head and neck located in the bone tissue. However, our qualitative literature analysis revealed that only a small number of articles fulfil a level of methodological quality that allows comparison of treatment effects.” (Clean Copy Line 417-420)

Having set the criteria for eligibility for the meta-analysis before conducting the initial literature research, we were hesitant to change these in the process. This ensures a transparent and robust research and analysis procedure. However, the heterogeneous reporting quality of the publications prompted the authors to add appropriate objectives to yield results that may be of interest to the reader and maintain a credible meta-analysis eligibility.

Therefore, the reviewer is correct when mentioning this change of objectives throughout the process of writing.

It appears, that the fact that the criteria for eligibility for meta-analysis were developed by the authors themselves has not been described adequately throughout the manuscript. As seen above, we modified these sections accordingly.

Note #2

In addition, various adapted risk of bias tools are available for non-RCT studies, these could be easily applied to the present review.

We thank the reviewer for this remark. Risk of bias assessment is a crucial aspect in the evaluation of the quality of scientific literature and we have long discussed this issue during the process. As mentioned in the methods section lines 148-151 we found utilization of risk of bias tools obsolete for the identified articles as none of them compare different interventions. In line 148 to 151 we have outlined the possible biases that may be relevant in the literature. To date there is no appropriate tool for the assessment of the risk of bias available. In 2018 Bero et al. analyzed a tool for studies of exposure (ROBINS-E) but have identified practical concerns with it and concluded that it does not meet the standard for evaluation. (The risk of bias in observational studies of exposures (ROBINS-E) tool: concerns arising from application to observational studies of exposures. DOI: 10.1186/s13643-018-0915-2

Note #3

Although the writers are critical and aware of the potential biases in the selected articles and opted to critically appraise in Table S2, I believe among others study design and sample size should be included (e.g. assessing the papers on directness of evidence and risk of bias).

We would like to thank the reviewer for this remark. We did include details on all evaluated articles in Table S5 which can be found in the appendix. The amount of detailed information in that table was not deemed practical to be included in the manuscript as interested readers can easily access this additional information.

Note #4

At this point the review lacks an easily assessable overview (i.e. flowchart or supporting figure) of the various treatments performed and its outcome in terms of for example success or complication rate, which I believe is the main objective of the review. Plenty of published reviews have coped with heterogeneity in results and alternatively analyzed (e.g. semi-quantitative) and visualized these results in various supporting manners.

The reviewer rightfully points out another important aspect. A clear and simple overview of applied methods and reported results such as outcome and complications is a desirable asset to the readability of a review. After discussing the results of our review, the authors had to acknowledge the vast heterogeneity of the articles in terms of applied therapy, reported characteristics of the study groups and display of results. As this heterogeneity is one of the major findings of the review the authors decided not to design such an overview to prevent the suggestion of comparability of articles. The methodological shortcomings of the reviewed articles unfortunately prohibit such an overview.

Note #5

I would also strongly advise to re-write the paper with a native English speaker as the paper contains multiple errors which decreases the joy of reading.

We would like to thank the reviewer to bring up this issue. We have corrected the manuscript with the help of a native speaker to improve the readability of the text and enhance the joy of reading. All changes are traceable in the markup version attached. However, changes in formatting are not documented.

Abstract

Note #6

Objective: I believe the secondary aim is obsolete as a methodological assessment of the selected papers is one of the key elements of performing a systematic review.

We thank the reviewer to point out this inaccuracy. The section was corrected accordingly. Please see reviewers note #1.

Note #7

Methods: “Criteria for meta-analysis were implemented.” I think you are meaning to say that a meta-analysis was performed?

The reviewer brings out an important point. With the background of existing literature and clinical experience, the authors have agreed on criteria for articles to be eligible for meta-analysis. Those criteria are referred to in the above sentence. We thankfully corrected this confusing phrase and hope to improve comprehension.

We changed the following paragraph in Methods in the Abstract section:

“The literature search was conducted on PubMed, Embase, the Cochrane Library, and scholar.google.com. Studies, meeting the acceptable reference standard underwent meta-analysis. All identified literature underwent methodological quality analysis.” (Clean Copy Line 31-33)

Note #8

Results: “Outcome analysis 38 varied with 45% of the authors using imaging parameters and 77% indicating the span of 39 follow-up of their entire patient collective” > this sentence is very vague for a new reader who was not involved in the research process, please re-write.

We thank the reviewer for pointing out the vague phrasing. We have re-written the section in Results in the Abstract section:

“Outcome analysis varied throughout. 45% of the authors used radiographic imaging for follow-up. 77% specified the span of follow-up of their entire patient collective.” (Clean Copy Line 38-40)

Note #9

Conclusion: This conclusion is not clear from the results section, please align these paragraphs.

We would like to thank the reviewer for highlighting this issue of confusion. Results of meta-analysis and the methodological quality should not be confused and should not imply unrepresentative results. We have aligned these sections and added details where appropriate.

We changed the following paragraph in Conclusion in the Abstract section:

“Our literature review revealed an absence of treatment or reporting standards for extracranial AVM of the head and neck. The meta-analysis is comprised of two articles and methodological quality is heterogeneous. We recommend implementing consistent reporting standards to facilitate comparability of studies and to provide robust data for the development of an evidence-based treatment strategy.” (Clean Copy Line 46-50)

We changed the following paragraphs in the Discussion:

“A major finding of our analysis was that therapeutic approaches for the management of AVMs vary considerably, not yielding treatment or reporting standards.” (Clean Copy Line 303-304)

“Given the low number of included cases, the varying reporting standards, and the heterogenous cohorts, the results of our qualitative analysis of the methodology must be interpreted with great care.” (Clean Copy Line 324-326)

Note #10

Advances in knowledge: Please add “data on AVMs /reporting standards addressing treatment of AVMs”, in this manner it seems you are referring to all data available. 

We have thankfully corrected this confusing phrase and added the necessary information.

We changed the following phrasing in the Advances in knowledge in the Abstract:

“Our analysis demonstrated that the published data on extracranial AVMs of the head and neck is lacking in consistency and quality, prompting agreement for the need of standardized reporting on AVM treatments.” (Clean Copy Lines 54-56)

Note #11

Introduction:

Lines 69 -70: This objective is different of the objective in the abstract, even though outcome is a consequence of a treatment, you should define what is “most effective treatment” ? What outcome are you interested in?

We thank the reviewer for showing this poor phrasing and have specified the outcome.

We changed the following section in the Introduction:

“In this review we focused on the analysis of articles covering the treatment of extracranial AVMs of the head and neck with the use of sclerotherapy and embolization.” (Clean Copy Line 74-75)

Note #12

Lines 70-71: See previous comment: evaluating the methodological quality of these papers is part of the systematic review.

See previous correction. We changed the following paragraph in the Introduction:

“Prior to the initial research, we developed the inclusion criteria for eligibility for meta-analysis (or acceptable reference standard). Methodological quality standards for analyzing the remaining articles were developed during the review process.” (Clean Copy Line 75-78)

Note #13

Line 73: Why are you only addressing the papers from 2000? Perhaps due to interventional developments? Please add your rationale.

We thank the reviewer for asking this important question. We have indeed considered this time frame since development of interventional radiology has been most remarkable during that time. We have thankfully added the rational in the methods section.

We changed the following section in the Methods:

“This studied timeline marks a rapid progression in the evolution of interventional radiology.” (Clean Copy Line 102-103)

Note #14

Methods

Lines 90-92: Consider to only mention your search in August 2020, it is OK to update your search once it gets outdated during the writing process, though only if it is performed in a similar fashion. I would recommend to add a second reviewer in the primary selection of eligible articles (or a substantial percentage of the eligible articles)

This sentence was phrased in a confusing manner. We did not perform additional research during the process of the review. Instead, we have added articles found in quotes and citations. This was done to ensure that results from the entirety of existing literature was analysed. The applied search string and keywords might have been missing vital aspects of the topic. By doing so we tried to reduce this possibility to a minimum.

We have rephrased the sentence from the Methods section to reduce the misunderstanding.

“Further publications identified in citations and quotes throughout the initial literature review were included in scope and underwent the same review process, with the last addition made in August 2020.” (Clean Copy Line 98-100)

Note #15

Lines 112-113: In this statement you are narrowing your research question and this should be used throughout your manuscript.

We thank the reviewer for this comment and have used the narrowing of the research question throughout the manuscript. 

Note #16

Line 123: To my opinion comparison in the PICO structure should be the other invention group you are comparing with, so that would be any primary treatment other than “sclerotherapy and embolization” or any patient in need of a secondary treatment and so on. At this point the paragraph after “Comparison” is just a brief sentence of your “Outcome” paragraph and does not give any additional information.

We thank the reviewer for the remark that correctly points out the rationale for the comparison of treatment options. One of the findings of our work was that to date there is no treatment standard for extracranial AVM and no superior treatment option has been identified. A comparison to an established treatment would make sense, as has been stated by the reviewer. The authors have opted to aim to identify a superior treatment option within the already implemented options. This information was humbly put in the manuscript.

We added the following paragraph in the Comparison section in the Methods:

“Leveraging meta-analysis, we compared radiological outcome and complication rates of the different treatment options for extracranial AVMs with the goal of identifying a superior treatment method within the existing literature.” (Clean Copy Line 132-134)

Note #17

Line 142: The authors should seek to another approach to adequately assess directness of evidence and risk of bias in the selected articles.

As mentioned in note #2 the authors have planned to implement risk of bias tool before conducting the initial research. It was deemed obsolete to implement these tools for all articles, but the authors indeed took up this suggestion for articles that were meta-analysed and used the ROBINS-I tool. The directness of evidence of all articles is outlined in the manuscript in the result section. As mentioned above, the authors have refrained from further assessment as results may yield wrongly suggestive results. 

Note #18

They could also consider to discuss the case reports separately from the other articles with a sample size of e.g. >15 or >20 patients. 

This suggestion by the reviewer is valuable for the discussion of the setup of the review. The authors designed the review without a limit to the sample size in order not to miss data and results. Considering the low incidence of extracranial AVM of the head and neck, results from any published treatment option may be of benefit. Thus, publication bias for the analysed literature can be limited. Furthermore, if the authors altered study groups or developed sub-groups in the process, results that may wrongfully suggest superior therapies can emerge. The authors tried to prevent this effect by purposefully leaving the sample size unrestricted.

Results

Note #19

Line 252: “four articles reported either mean of maximum follow-up time only” > if this FU was longer than 1 year I guess these articles could still be included (as is stated in the methods). If so, please add why these papers were excluded.

We would like to thank the reviewer for this comment. In designing the study, the authors agreed to limit the minimum time of follow-up to one year. This means that all included individuals from the study groups must have a minimum follow-up time of one year or 360 days. If articles only stated, the mean follow-up time there is a possibility that not all individuals have had that minimum requirement. The same applies for the maximum follow-up time. The authors do agree with the reviewer that the phrasing was unclear and have corrected the sentence in the Quality of follow-up section in Methods:

“For cerebral AVM, a minimum follow-up period of one year is recommended to detect lesion recurrence after therapy.(9) Analogously, a follow-up time of at least one year was considered necessary for inclusion in our meta-analysis. As mean or median time of follow-up of the study group (as well as maximum) time do not yield information on the individual minimum follow-up time, that data alone was considered insufficient, and the related literature was excluded from analysis.” (Clean Copy Line 180-185)

Note #20

For a meta-analysis, Pearson Chi square and Mann Whitney U tests are not appropriate. 

This is an important remark that we implemented in the manuscript. We have now applied Chi-square for the calculation of a significance in difference. In general, we would have liked to apply regression tables to perform a more precise calculation in the meta-analysis and visualization with forest-plots. Since we could only identify two studies to undergo these calculations, we decided to use Chi square as straight forward and solid approach for this statistical issue. As this does indeed show a limitation of the quantitative analysis, we gave explanation in the limitation section.

Discussion

Note #21

Another goal “representative meta-analysis” is added for the first time in the discussion part. The objectives should be similar throughout the manuscript.

We thank the reviewer for pointing this issue out. We have revised all sections of the manuscript regarding the phrasing and details on the objectives of the study. Particularly the discrimination between the implementation of the acceptable reference standards, the conducted meta-analysis, and the qualitative analysis of the methodology of the articles was verified and corrections were made.

Note #22

The discussion shows that the authors do have substantial knowledge of the subject as they are able to discuss and reflect on the different treatment options. It would be of interest if the authors could add their view on what the suggested reporting standards should contain or focus on.

We appreciate the reviewer’s interest in the authors’ suggestions for reporting standards. To emphasize the subjective character of these suggestions the authors added four aspects for future reporting standards at the end of the Discussion and in the Conclusion. 

Note #23

Line 367-369: To my opinion a long list of Author et al is not reader-friendly, please re-write.

We thankfully received the suggestions and changed the section in Discussion accordingly.

“Lastly, varying outcome assessments and definitions make comparisons difficult. For instance, a devascularization above 90% is considered “complete” by three authors, but only considered as “partial” by nine others (Table 4).” (Clean Copy Line 383-385)

Conclusion

Note #24

All recommendations for reporting standards should be placed in the discussion. In addition please explain why >99% (why not 90%?)a devascularisation is used and how this should be measured.

Regarding suggestions for reporting standards, please see note #20 above.

For radiographic extinction of an AVM lesion, proof of complete devascularization is necessary. When suggesting a 90% devascularization as adequate, the reader may come to believe that the status “cure” may also be obtained with the detection of a residual nidus. This can lead to misunderstanding when looking at the results of the qualitative analysis of the articles, where “cure” is a vague term. Therefore, the nidus must not be detectable in DSA while leaving the chance of missing that 1% of contrast volume with the naked eye in DSA.

Note #24

Also the strengths and limitations should be placed in the discussion.

We thank the reviewer for this valuable point and have changed the sections accordingly.

Note #25

Fig 1 PRISMA flowchart

- Please take note of the “simple” PRISMA set up of the flowchart in other published systematic reviews. This flowchart is quite chaotic and hard to follow for an interested reader.

This is a very helpful suggestion by the reviewer and has helped us to redesign the graph in hopes to show a clearer picture of the research process. (Please see attached Fig1_Revised)

Supporting data:

Note #26

Please add all supplemental data to one document to help the reader find the information they need to comprehend the message of your paper

When submitting the manuscript and the supporting data, we held on to the submission guidelines outlined on the PLOS webpage. If there is need to change the layout or order of the submitted data, we are more than happy to do so.

Note #27

It would be better to give an summary score / rating to the separate articles in the table S2 in order to see which articles are the most appropriate to answer your research question. Also the table should stand on its own, meaning the table should be informative enough to fully understand independently of the paper. In my opinion the legend is too brief for a reader which is not as familiar with the subject as the writers.

This comment is very helpful to recognize the importance of clear and brief overviews and has been a point of discussion for the authors in the past, as seen in Note #4. A scoring system or any other form of ranking of the articles may suggest to the reader that reporting standards were maintained by the authors of the literature to a varying extend. Yet, there are no standards, and such ranking may wrongfully imply the superiority of the quality of one article over another. Therefore, we would rather not consider such a method at this point. If, however, the application of other symbols or coloring were of help for readability, we would be happy to improve the table in the desired way.

Note #28

I understand the length of the Table S5 makes it unfavorable to add in the main body of the text, but as this table may be the most important finding of the study I would recommend the writers to think of another way to add this information to the main manuscript. Perhaps a clean flowchart with roughly categorized treatments and their success or complication rate would be of interest.

This important issue has been pointed out by the review and is addressed in note #4 and #27. We would like to refer to the answers there. However, we are happy to consider changes in this regard if the reviewer still deems this of utmost importance.

---

## [Decision Letter · Decision Letter 1]

10 May 2022

Interventional therapy of extracranial arteriovenous malformations of the head and neck – A systematic review

PONE-D-21-11556R1

Dear Dr. Nikoubashman,

We’re pleased to inform you that your manuscript has been judged scientifically suitable for publication and will be formally accepted for publication once it meets all outstanding technical requirements.

In your final version please add an example figure as outlined by reviewer 1 of an AVM pre and post embolization!

Kind regards,

Stephan Meckel, MD, PhD

Academic Editor

PLOS ONE

Additional Editor Comments (optional):

Reviewers' comments:

Reviewer's Responses to Questions

**Comments to the Author**

1. If the authors have adequately addressed your comments raised in a previous round of review and you feel that this manuscript is now acceptable for publication, you may indicate that here to bypass the “Comments to the Author” section, enter your conflict of interest statement in the “Confidential to Editor” section, and submit your "Accept" recommendation.

Reviewer #1: All comments have been addressed

2. Is the manuscript technically sound, and do the data support the conclusions?

Reviewer #1: Yes

3. Has the statistical analysis been performed appropriately and rigorously? 

Reviewer #1: Yes

4. Have the authors made all data underlying the findings in their manuscript fully available?

Reviewer #1: Yes

5. Is the manuscript presented in an intelligible fashion and written in standard English?

Reviewer #1: Yes

6. Review Comments to the Author

Reviewer #1: All comments have been successfully addressed and I am pleased with the revised manuscript and supportive rationale. One last remark as cherry on top of the manuscript; it would be interesting to add an example figure of i.e. DSA before and after treatment of the AVM. Best wishes in future research

7. PLOS authors have the option to publish the peer review history of their article (what does this mean?). If published, this will include your full peer review and any attached files.

Reviewer #1: **Yes: **Constance JHCM van Laarhoven, MD PhD

---

## [Editor Report · Acceptance letter]

20 May 2022

PONE-D-21-11556R1 

Interventional therapy of extracranial arteriovenous malformations of the head and neck – A systematic review 

Dear Dr. Nikoubashman:

I'm pleased to inform you that your manuscript has been deemed suitable for publication in PLOS ONE. Congratulations! Your manuscript is now with our production department. 

Kind regards, 

on behalf of

Prof. Dr. Stephan Meckel 

Academic Editor

PLOS ONE